# Impacts of resistance training combined with vibration training on the IGF-1/PI3K/AKT/FOXO3 axis and clinical outcomes in patients with sarcopenia: A protocol for a randomized controlled trial

**Haoyang Zhou**[1], **Jinfeng Yang**[1], **Na Li**[2], **Jinying Li**[1], **Jianxin Ran**[1], **Yan Zheng**[1], **Yifan Long**[1], **Fang Cheng**[1], **Yuanpeng Liao**[3]*

**1** Department of Sports Medicine and Health, Chengdu Sport University, Chengdu, China, **2** The National Clinical Research Center for Geriatrics, West China Hospital of Sichuan University, Chengdu, China, **3** Affiliated Sport Hospital of Chengdu Sport University, Chengdu, China

* liaoyuanpeng@cdsu.edu.cn

## Abstract

### Background

Sarcopenia is an age-associated disorder characterized by a progressive decline in skeletal muscle mass, strength, and physical function. The condition is linked to low levels of anabolic hormones such as insulin-like growth factor 1 (IGF-1), with its downstream phosphatidylinositol 3 kinase (PI3K)/ protein kinase B (AKT)/ forkhead box protein O3 (FOXO3) signaling pathway. There is growing evidence that resistance training (RT) or vibration training (VT) could improve physical functioning in individuals with sarcopenia. However, the related physiological influence of exercise on sarcopenia remains elusive.

### Method

This prospective randomized controlled trial will be conducted among 96 participants, aged between 65 and 80 years. In participants, sarcopenia diagnosis will be confirmed based on the Asian Working Group for Sarcopenia criteria, and participants will be randomized into either control, RT, VT, or RVT (combined RT and VT) groups. The intervention will last 12 weeks, with assessments performed at baseline, 12 weeks (after intervention), and 24 weeks (follow-up). The primary outcomes will include skeletal muscle mass, handgrip strength, and gait speed. Secondary outcomes comprise IGF-1 concentrations, PI3K/AKT and FOXO3 protein activity, quality of life, and timed-up-and-go test performance assessments.

**Data availability statement:** No datasets were generated or analysed during the current study. All relevant data from this study will be made available upon study completion.

**Funding:** The Institute of Sports Medicine and Health at Chengdu Sports University and the Basic Cultivation Project of Sichuan Provincial Expert Workstation (Grant # SCZJJCC-16) have funded this clinical study.

**Competing interests:** The authors have declared that no competing interests exist.

## Discussion

This clinical study aims to elucidate the potential modulation of molecular mechanisms in vivo for combined RT and VT in sarcopenia patients and to identify the effects of the intervention on physical function.

## Trial registration

ChiCTR, ChiCTR2400083643. Registered on April 29, 2024.

## Introduction

Sarcopenia is an age-related disorder characterized by a progressive decline in skeletal muscle mass, strength, and physical function [1]. In China, the overall prevalence of sarcopenia among community-dwelling older adults has reached 9.3%. According to the Asian Working Group for Sarcopenia (AWGS) criteria, the estimated prevalence is approximately 6.4% in men and 11.5% in women aged 60 years and above [2]. Individuals with sarcopenia could have accidental falls [3], fractures of the upper and lower limbs [4], physical disability [5], cognitive impairment [6], and poor quality of life [7], with an increased mortality risk [8]. The changes observed in older individuals with sarcopenia can have clinical and physiological impacts, such as age-related inflammation, metabolic abnormalities, and endocrine changes [9–11].

Sarcopenia has been linked to reduced levels of anabolic hormones, specifically insulin-like growth factor 1 (IGF-1), among individuals aged 60 years and above [12]. Forkhead box protein O3 (FOXO3) is a downstream component of the IGF-1 signaling pathway and is crucial for catabolic/anabolic protein homeostasis and protein breakdown within the musculoskeletal system [13]. IGF-1 activates phosphatidylinositol 3 kinase (PI3K) by binding to its receptor, which catalyzes the conversion of phosphatidylinositol-4,5-bisphosphate (PIP2), a membrane-bound molecule, to phosphatidylinositol-3,4,5-trisphosphate (PIP3), which recruits and activates protein kinase B (AKT). This, in turn, phosphorylates FOXO3 and inhibits its nuclear translocation and transcriptional activity [14]. In the anabolic state, this process decreases the expression of FOXO3's downstream target genes (e.g., MuRF-1 and Atrogin-1 act as ubiquitin ligases and induce proteasomal degradation) and inhibits muscle protein degradation [15]. When IGF-1 signaling is diminished (e.g., owing to aging or stress), AKT activity decreases and FOXO3 dephosphorylates and enters the nucleus, stimulating an increase in MuRF-1 and Atrogin-1 levels, as well as promoting muscle atrophy *via* the ubiquitin-proteasome pathway [16–18]. Thus, a lower IGF-1 level can lead to a higher FOXO3 activity via the PI3K/AKT pathway, causing protein degradation and skeletal muscle atrophy [19], thereby resulting in the onset of sarcopenia in older adults. Among the various regulatory genes in the musculoskeletal system, FOXO3 is crucial; therefore, its inhibition might be a potential strategy for treating muscle atrophy and sarcopenia [12]. On the basis of the above evidence, it is crucial to examine the influence of different sarcopenia treatment strategies on the IGF-1/PI3K/AKT/FOXO3 pathway.

Given the absence of effective medications to specifically treat sarcopenia [20], non-pharmacological treatment remains the most promising approach to substantially enhance physical health and muscle function in older sarcopenia patients [21]. Resistance training (RT) is a widely implemented intervention for older sarcopenic adults [22]. However, RT often has high-technology requirements [23,24] and can easily produce muscle fatigue in this population [25], leading to reduced compliance with RT, especially in those who are frail. Additionally, implementing RT in clinical settings often proves challenging. Vibration training (VT), which employs platforms generating variable frequencies and amplitudes, has emerged as a potential intervention for sarcopenia in older adults [26]. Recently, VT has been increasingly employed to delay the decline in muscle function in older sarcopenia population [27–29]. A meta-analysis revealed that VT could reverse reduced muscle size, strength, and physical performance [30]. Compared with RT, the advantages of VT include being easy to perform, having short session durations, and appropriately compensating for muscle fatigue from the RT session. Wei et al. demonstrated that VT could significantly improve quadriceps voluntary activation in older sarcopenia patients [28]. Zhu et al. confirmed that tai chi combined with VT could enhance the physical function, mass, and strength of the muscles in sarcopenic men [31]. Furthermore, VT's optimal frequency and duration can be adjusted to maximize benefits in increasing muscle size, strength, and physical performance in age-linked muscle loss patients [32,33]. Thus, combining VT with RT may offer a more effective approach for mitigating sarcopenia.

In terms of the cell signaling pathways, animal studies have shown IGF-1 pathway activation via VT [34]. Increased IGF-1 levels and improved muscle mass were also observed in sarcopenic patients after RT [35]. However, another study found no increase in inactivated FOXO3 in older women after 12 weeks of RT, implying limited benefits of RT in these women [36]. Preclinical studies have further demonstrated that vibration stimuli can promote muscle protein synthesis via mechanical loading–induced IGF-1 expression. For instance, Li et al. [34] reported that VT activated the PI3K/AKT/mTOR pathway and suppressed muscle atrophy in aged mice. Human studies have also reported increased serum IGF-1 levels following VT, suggesting systemic anabolic responses [28]. The proposed mechanism in this study is that VT enhances neuromuscular activation and mechanosensitive stimulation, which upregulates IGF-1 production. This in turn activates the PI3K/AKT signaling cascade, leading to phosphorylation of FOXO3 and reduced expression of its downstream catabolic targets, MuRF-1 and Atrogin-1. Thus, VT may shift the muscle environment toward an anabolic state through modulation of the IGF-1/PI3K/AKT/FOXO3 axis. This forms the central mechanistic hypothesis of the present protocol.

Sex-specific differences in sarcopenia are well recognized. A recent study has shown that women tend to experience a more rapid decline in muscle strength relative to muscle mass, whereas men are more likely to exhibit significant reductions in muscle mass with aging [37]. These disparities may be influenced by differences in hormonal status (e.g., estrogen vs. testosterone), muscle fiber composition, and neuromuscular control mechanisms [38]. Additionally, evidence suggests that men and women may respond differently to exercise interventions such as resistance and vibration training [39,40]. Therefore, sex-specific considerations are essential when designing sarcopenia-related interventions and interpreting outcomes.

In this study, we investigate the impact of RT combined with VT on the IGF-1/PI3K/AKT/FOXO3 axis in patients with sarcopenia. Using a single-blind randomized controlled trial, we will assess whether this combined exercise therapy exerts significant molecular and functional effects, while also exploring potential differences in treatment response between sexes.

## Methods and analysis

### Study design and grouping

This randomized controlled trial will identify the effects of RT combined with VT on the IGF-1/PI3K/AKT/FOXO3 axis in a population with sarcopenia. Participants will be screened for sarcopenia, as defined by the AWGS in 2019, and recruited based on eligibility criteria. Fig 1 summarizes the participant's enrollment process, interventions, and assessment schedule. Once informed consent is received, the participants will be randomly categorized into (1) control, (2) RT, (3) VT, and

| | STUDY PERIOD | | | | |
|---|---|---|---|---|---|
| | Enrolment | Allocation | Treatment period | | Fallow-up |
| TIMEPOINT** | -Week$_1$ | Week$_0$ | Week$_1$ | Week$_{12}$ | Week$_{24}$ |
| **ENROLMENT:** | | | | | |
| **Eligibility screen** | X | | | | |
| **Informed consent** | X | | | | |
| **Baseline assessment** | X | | | | |
| **Allocation** | | X | | | |
| **INTERVENTIONS:** | | | | | |
| RT | | | ●━━━━━━━━━━━━● | | |
| VT | | | ●━━━━━━━━━━━━● | | |
| RVT | | | ●━━━━━━━━━━━━● | | |
| Education (No intervention) | | | ●━━━━━━━━━━━━● | | |
| **ASSESSMENTS:** | | | | | |
| Hand grip | X | | | X | X |
| Gait speed | X | | | X | X |
| Skeletal muscle mass | X | | | X | X |
| IGF-1 | X | | | X | X |
| PI3K/ AKT | X | | | X | X |
| FOXO3 | X | | | X | X |
| Time-up-and-go | X | | | X | X |
| SF-36 | X | | | X | X |
| Participants' compliance | | | ●━━━━━━━━━━━━● | | |
| Adverse events | | | ●━━━━━━━━━━━━● | | |
| Dropout reasons | | | ●━━━━━━━━━━━━● | | |

**Fig 1. Enrollment, interventions, and assessments schedule.** RT: resistance training, VT: vibration training, RVT: resistance training combined with vibration training, IGF-1: insulin-like growth factor 1, PI3K/AKT: phosphatidylinositol 3-kinase/ protein kinase B, FOXO3: forkhead box protein O3, SF-36: 36-item short form health survey.

(4) combined RT and VT (RVT) groups. Intervention will last 12 weeks, with data collection and assessments at baseline, 12-week (after intervention), and 24-week (follow-up). The 12-week intervention duration was selected based on existing literature indicating that this time frame is sufficient to induce meaningful improvements in muscle mass, muscle strength, and molecular signaling among older adults. Prior studies on resistance and vibration training for sarcopenia typically range between 8–12 weeks, with 12 weeks allowing both early neural adaptation and later-stage hypertrophy to manifest [41–43]. The 24-week time point (i.e., 12weeks post-intervention) was included to evaluate the sustainability of training-induced benefits. Similar follow-up durations have been adopted in previous sarcopenia trials to assess the persistence or decay of functional gains after supervised training ends [44]. Muscle strength, mass, physical performance, FOXO3 protein expression levels, and quality of life will be assessed in all participants. Fig 2 depicts the study design, and S1 File contains the SPIRIT checklist.

## Sample size determination

The required sample size was calculated using G*Power (version 3.1.9.7, Germany) for a two-way repeated-measures ANOVA (within-between interaction), with a significance level of $\alpha = 0.05$, power $(1–\beta) = 0.80$, and four intervention groups (Control, RT, VT, RVT). The effect size was set to $f = 0.40$ (partial $\eta^2 = 0.14$), which reflects a moderate-to-large effect based on prior studies assessing changes in IGF-1 expression and physical function following resistance and vibration training in older adults. The primary endpoint for power estimation was the change from baseline to 12 weeks, representing the completion of the supervised intervention. The 24-week point is considered a follow-up to assess sustainability but was

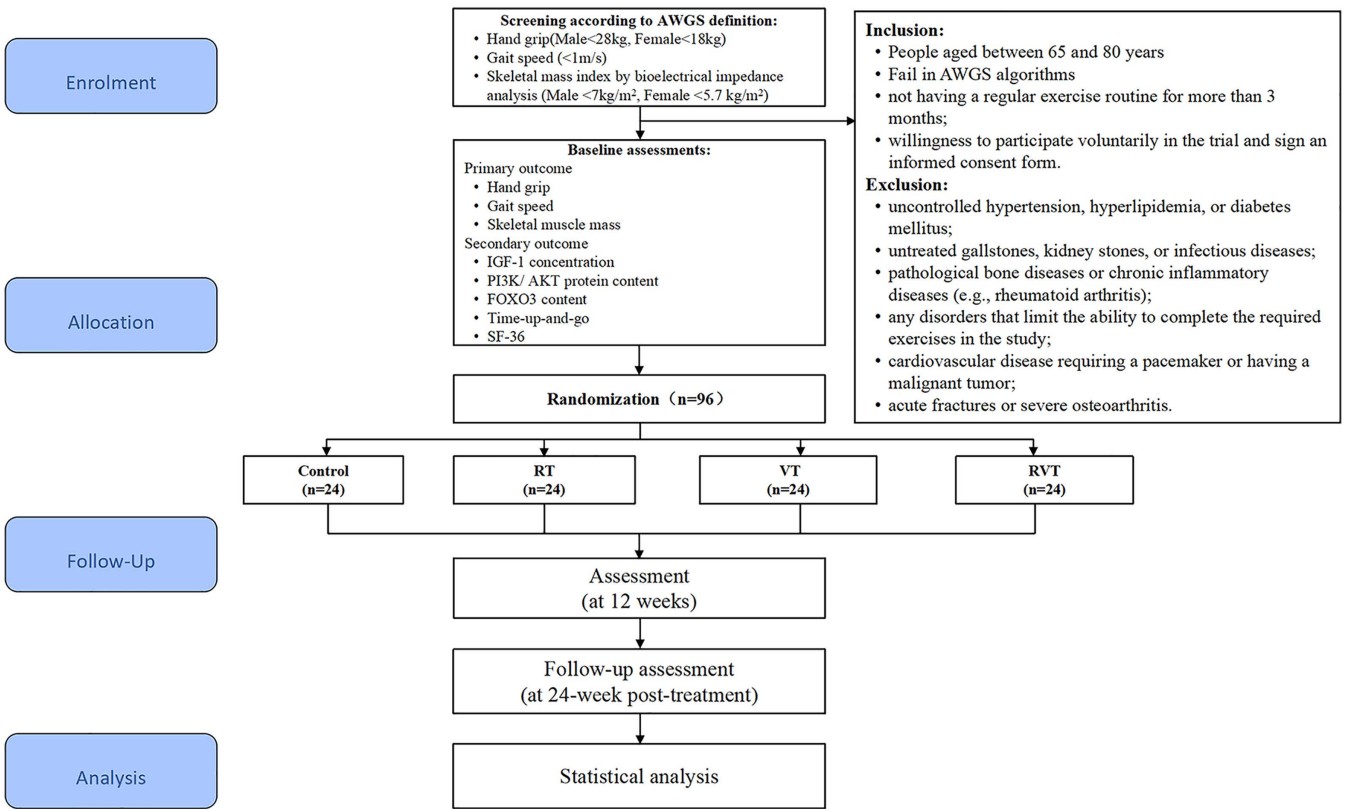

**Fig 2. Study flow chart.** RT: resistance training, VT: vibration training, RVT: resistance training combined with vibration training, IGF-1: insulin-like growth factor 1, PI3K/AKT: phosphatidylinositol 3-kinase/ protein kinase B, FOXO3: forkhead box protein O3, SF-36: 36-item short form health survey.

not included in the power calculation. Considering a 20% dropout rate, the minimum required sample size was estimated at 73 participants [45]. Therefore, we plan to recruit 96 individuals (24 per group) to ensure adequate statistical power. In view of multiple primary outcomes and six pairwise comparisons among four groups, Holm–Bonferroni correction will be applied to adjust for potential inflation of type I error. Additionally, the hypothesis is based on additive rather than synergistic effects between RT and VT. Hence, interaction effects were not included in the sample size model but will be explored post hoc.

## Inclusion criteria

The inclusion criteria are as follows: 1) individuals between age 65 and 80 years; 2) sarcopenia diagnosis based on the AWGS 2019 criteria: appendicular skeletal muscle mass index < 7.0 and < 5.7 kg/m$^2$ for men and women, respectively, using a bioelectrical impedance analyzer, handgrip strength < 28 and < 18 kg in men and women, respectively, and/or gait speed < 1.0 m/s in the 6-meter walk test; 3) not regularly exercising for > 3 months; 4) willingness to participate voluntarily and sign the informed consent form for this trial.

## Exclusion criteria

The exclusion parameters included individuals with: 1) uncontrolled hypertension, hyperlipidemia, or diabetes mellitus; 2) untreated gallstones, kidney stones, or infectious diseases; 3) pathological bone diseases or chronic inflammatory diseases (e.g., rheumatoid arthritis); 4) any disorders that limit the ability to complete the required exercises in the study; 5) cardiovascular disease requiring a pacemaker or having a malignant tumor; 6) acute fractures or severe osteoarthritis; 7) current use of medications known to affect muscle metabolism or hormonal balance, including but not limited to systemic corticosteroids, anabolic steroids, or hormone replacement therapy. Additionally, all participants' current use of medications for cardiovascular or metabolic conditions (e.g., beta-blockers, statins, insulin sensitizers) will be documented at baseline. If imbalances between groups are identified, these variables will be considered as covariates in the statistical analysis; 8) current or recent (within the past 3 months) use of nutritional supplements that may affect muscle metabolism or sarcopenia progression, such as protein powders, amino acids, vitamin D, or calcium supplements.

## Recruitment strategy

Potential participants will be recruited through educational seminars at partner community centers, promotional materials distributed both onsite and online, and referrals from collaborating organizations. Screening will investigate handgrip strength, bioelectrical impedance, and gait speed based on the AWGS 2019 criteria. Eligible individuals will receive invitations to participate in the study at Sichuan Province's Sports Medicine Key Laboratory. Prior to enrollment and during the informed consent process, all participants will be informed about the study procedures, as well as the potential risks and benefits. Participants will retain the right to withdraw unconditionally at any stage of the research.

## Randomization and blinding

After acquiring the signed consent forms, 96 participants will be screened for eligibility and will be randomly categorized into the RT, VT, RVT, or control group, at a 1: 1: 1: 1 ratio using computer-generated simple randomization. A reproducible random number sequence will be generated using IBM SPSS (version 27.0), and participants will be sorted in ascending order based on these numbers. Group allocation will be determined using a modulo-4 method: remainder 0 = RT, remainder 1 = VT, remainder 2 = RVT, and remainder 3 = control, ensuring equal group sizes. No baseline characteristics (e.g., age, sex) will be used as stratification factors during randomization. To minimize allocation bias, group assignments will be concealed using sequentially numbered, opaque, sealed envelopes (SNOSE), and only personnel responsible for randomization will have access to the sequence. Investigators, outcome assessors, and statisticians will remain blinded to group allocation throughout the study.

## Interventions

**Group protocols.** Participants in the RT, VT, and RVT groups will undergo pre-training to acquaint them with the respective protocols before commencing formal training. Before and after each treatment session, participants in the three intervention groups will complete a 5-minute warm-up and cool-down. All training sessions will be conducted in a group under the guidance of professional physiotherapists. Throughout the training period, participant attendance and adherence will be systematically documented. If a participant cannot start the training session on time for any reason, they can join training sessions that will be carried out at other times during that week. All participants will be asked to refrain from engaging in any additional physical activity and maintain their usual daily habits throughout the study.

**Resistance training program.** Participants assigned to the RT group will undergo a 12-week exercise program comprising 3 weekly sessions. Before the intervention begins, all participants will perform a one-repetition maximum (1RM) test to elucidate baseline muscle strength and determine individualized training intensities. The 1RM will be estimated using a submaximal testing approach, in which participants perform 3–5 repetitions at approximately 60%–80% of their perceived maximum load for selected TheraBand-based resistance exercises (e.g., knee extension, hip abduction, elbow flexion). The Brzycki formula (estimated 1RM (kg) = submaximal weight (kg)/ $(1.0278 - 0.0278 \times$ maximal number of repetitions)) [46] will be applied to estimate 1RM values from these submaximal repetitions. All tests will be supervised by experienced physiotherapists, and standardized rest intervals of 1 minute will be maintained between sets to minimize fatigue. Based on the estimated 1RM, participants will be assigned an appropriate TheraBand (color-coded by resistance level: yellow, red, green, blue, black) and instructed to maintain a specific elongation length (e.g., 100% or 200%) to approximate the desired resistance. This allows adjustment of load to approximately 70% of 1RM, consistent with elastic resistance training literature [47,48]. To enhance precision, perceived exertion will be concurrently monitored using the Borg CR-10 scale [49,50], ensuring training load consistency throughout the intervention. The training intensity will be set at 70% of the 1RM for the entire 12-week program. Each training session will comprise 3 phases: a 5-minute warm-up, the main 30-minute RT exercises, and a 5-minute cool-down. During the 5-minute warm-up, participants will mainly perform neck, shoulder, lower back, hip, knee, and ankle stretches. Participants will perform three sets of 10 repetitions for each exercise, with a 1-minute rest between sets. The exercises will be performed using a TheraBand to reduce the stress of free weights on participants' joints and increase their motivation to engage in RT [51,52]. RT exercises will focus on lower limb muscles (knee flexion/extension, hip abduction), upper limb muscles (shoulder extension/abduction, elbow flexion/ extension), and core muscles (including chest press). Following these exercises, participants will perform a 5-minute cool-down protocol matching the warm-up in structure.

**Vibration training program.** The VT cohort will receive VT sessions 3 times/week for 12 weeks. VT will be performed using a vibrating platform (Pro5 AIRdaptive; Power Plate, USA) that produces vertical vibrations. Each session will last approximately 19 minutes, comprising a 5-minute warm-up, a 9-minute vibration period, and a 5-minute cool-down. The vibration frequency will gradually increase to 40 Hz with a peak-to-peak amplitude of 4 mm. Participants will stand barefoot on the vibrating platform during VT, maintaining a 60° knee flexion while grasping the vibrating rope with both hands for balance. The VT will be maintained for 1.5 minutes and repeated for four sets each day [33], with a 1-minute rest between each set.

**Combined RT and VT program.** The RVT group will receive a 12-week RT combined with a VT program thrice weekly. The intensity of RT in the RVT group will be adjusted based on each participant's 1RM test results. The intensity will be set at 50% of 1RM for the first 4 weeks, increased to 55% during weeks 5–8, and further increased to 60% during weeks 9–12. This progressive design was adopted to improve tolerance and adherence, as participants in the RVT group receive an additional neuromuscular stimulus from vibration training. A gradual intensity progression has been shown to enhance safety and minimize musculoskeletal discomfort in older adults when initiating vibration-based exercise interventions. This approach is supported by previous studies [53,54], which recommended lower initial intensities in vibration training to allow neuromuscular adaptation In contrast, the RT group, which does not receive vibration

stimulation, follows standard sarcopenia guidelines and begins directly at 70% 1RM. Each workout will follow the same structure as the RT group, including warm-up, RT, and cool-down. Each RT session will be followed by an additional VT session, with the same parameters as those of the VT group.

**Educational courses.** Participants in the control group will attend a 60-minute education session twice monthly. Invited experts will deliver these sessions, covering topics such as aging, general health, and sarcopenia—including its definition, etiology, pathophysiology, clinical manifestations, and adverse effects in older adults. These educational courses are intended primarily to maintain participant engagement and minimize dropout. The sessions are standardized, lecture-based, and non-interactive, and do not include individualized exercise prescriptions or dietary counseling. Although it cannot be entirely excluded that exposure to health information may raise awareness or induce minor behavioral changes, previous studies have shown that education-only interventions in older adults generally do not lead to significant improvements in muscle strength, skeletal muscle mass, or physical function [55,56]. Therefore, the risk of overestimating the effects of exercise-based interventions due to this educational component is considered minimal.

## Outcome measures

**Primary outcomes.** The primary outcomes of this study include skeletal muscle mass, muscle strength, and physical performance. Skeletal muscle mass will be measured using dual-energy X-ray absorptiometry (DXA) through whole-body scanning. Total appendicular skeletal muscle mass will be calculated by summing the values from all four limbs. A threshold of < 7.0 kg for men and < 5.4 kg for women will be used to identify sarcopenia.

Muscle strength will be assessed using a calibrated handgrip dynamometer (5030JI, Jamar, IL, USA). Participants will be instructed to squeeze the device maximally for a few seconds while holding it at a 90° elbow angle by their side. Three measurements will be taken bilaterally, and the highest value will be recorded. According to the AWGS 2019 criteria, handgrip strength < 28 kg for men and < 18 kg for women indicates low muscle strength. Physical performance will be evaluated via the 6-meter gait speed test, with a cutoff of < 1.0 m/s used to define poor performance, based on the AWGS 2019 recommendations.

**Secondary outcomes.** IGF-1 concentrations will be assessed *via* an enzyme-linked immunosorbent assay (ELISA) kit (IGF-1 600 ELISA; DRG, Germany). Blood will be collected from the antecubital vein in a fasting state in serum separator tubes, centrifuged for 15 minutes at 3000 × g and 4°C to harvest serum, which will be stored at −80°C for biochemical analysis [57].

The PI3K/AKT signaling pathway and FOXO3 protein activity will be assessed using immunoblotting (Western blot) and cell isolation assay on specimens obtained via muscle biopsy. Muscle biopsies will be collected at three time points: baseline (pre-intervention), 12 weeks (post-intervention), and 24 weeks (follow-up), enabling longitudinal analysis of the IGF-1/PI3K/AKT/FOXO3 axis. Although more frequent sampling could provide a finer-grained understanding of molecular responses, ethical and practical concerns regarding repeated invasive procedures in older adults limit the feasibility of acute or intra-intervention biopsies in this study. The percutaneous muscle biopsy technique will be employed for lateral femoral muscle biopsy. Briefly, after 2–3 mL of 1% dimethocaine local anesthesia, aspiration will be performed. Then, about 50 mg of muscle tissue lacking blood vessels and connective tissues will be dissected. The samples will be immediately placed in cryotubes, frozen in liquid nitrogen, and stored at −80°C [58].

Then, the protein content will be evaluated *via* the bicinchoninic acid assay (BCA) method (Pierce, Appleton, WI, USA). Subsequently, equal amounts of protein will be subjected to 12% SDS-PAGE gel electrophoresis and transferred onto PVDF membranes, which will then be treated at 4°C with the primary antibodies against PI3k (ab140307,1:1000, Abcam, UK), Akt (SAB4500797, 1:1000, Sigma, USA), Akt (phospho T308) (ab38449, 1:1000, Abcam, UK), and FOXO3 (SAB2107951, 1:1000, Sigma, USA) overnight. Next, the membranes will be flushed with phosphate-buffered saline with tween (PBST), probed with the HRP-linked specific goat anti-rabbit IgG secondary antibody (ab150077, 1:3000, Abcam, UK) for 2 h at ambient temperature, and then washed thrice with PBST. The bands will subsequently be quantified *via* the

Image Lab system (Bio-Rad, Shanghai, China) [59,60]. The above analyses will be performed in the Sports Medicine Key Laboratory of Sichuan Province.

A timed-up-and-go test will be performed, during which the time it takes for participants to get up from the chair, walk 3 meters, turn around, go back to the chair, and then sit down will be recorded.

The validated Chinese version of the 36-item short form health survey will be employed to elucidate health-related quality of life. This instrument will evaluate both physical and mental components, with the total score reflecting overall quality of life, with higher values corresponding to better outcomes. [61]

## Statistical analysis

IBM SPSS v.20.0 will be utilized to analyze all the data. The primary outcomes will be assessed using two-way repeated-measures analysis of variance (ANOVA), with factors including time (baseline, 12 weeks, 24 weeks) and group (control, RT, VT, RVT). This approach was selected to allow direct estimation of group-by-time effects for predefined primary endpoints under a balanced design. Holm–Bonferroni adjustments will be applied for post-hoc pairwise comparisons. Missing data will be imputed via multiple imputation. Sex, along with baseline IGF-1, age, and muscle mass, will be included as covariates in analysis of covariance (ANCOVA) models to adjust for potential confounding effects. Effect sizes (partial $\eta^2$ for ANOVA; for pairwise comparisons, the Cohen's d with 95% confidence intervals were employed) will be categorized as small ($\eta^2 \geq 0.01$, $d \geq 0.2$), medium ($\eta^2 \geq 0.06$, $d \geq 0.5$), or large ($\eta^2 \geq 0.14$, $d \geq 0.8$), accompanied by exact p-values. The 95% confidence intervals for Cohen's d and model-based estimates will be calculated using either model-derived standard errors or the bias-corrected and accelerated (BCa) bootstrap method with 1,000 resamples, depending on distributional assumptions. To better accommodate repeated measures and potential within-subject correlations in secondary longitudinal outcomes, linear mixed-effects models will be used, incorporating random intercepts for participants and fixed effects for time, group, and their interaction. In addition, exploratory subgroup analyses stratified by sex will be conducted to evaluate potential sex-specific intervention effects. Subgroup interaction terms will be tested where appropriate, with $p < 0.10$ considered indicative of potential moderation. Statistical significance will be defined as $p < 0.05$ for all primary tests. Although the current sample size is not specifically powered to detect sex-based interaction effects, these exploratory analyses will inform the design of future sex-specific trials. All analyses will follow the intention-to-treat (ITT) principle. All randomized participants will be analyzed according to their assigned groups, regardless of protocol adherence or study completion status.

## Data collection and quality control

Before randomization, the characteristics of all the participants will be acquired during baseline evaluations. No data or information will be shared with third parties. Each participant will receive a unique identifier to ensure anonymity, and only personnel conducting statistical analyses will have access to the study database.

All investigators will undergo a comprehensive training program covering the intervention methods for each group, eligibility criteria, trial protocols, outcome measures, and accurate data recording. All practitioners will be certified rehabilitation therapists with at least 3 years of clinical experience in sports rehabilitation.

To increase adherence, participants will receive timely visit reminders and will incur no costs for assessments or treatments. Those who complete all study procedures will receive financial compensation.

## Assessment of adverse events and adherence

Adverse events occurring during the intervention will be systematically documented. Participants experiencing such events may withdraw from the trial at their discretion. Participant engagement with and adherence to the intervention will be assessed for each group. If a participant withdraws from the program, we will record the reason for withdrawal.

## Patient and public involvement

Participants will not contribute to the study design, outcome selection, or result interpretation, nor will they participate in drafting or revising the manuscript.

## Ethics and dissemination

The Ethics Committee of Chengdu Sport University approved this study on 23 April 2024 (approval no. 202466; S2-S3 File), confirming that it adheres to the Helsinki Declaration. Prior to participation, all participants will provide their written informed consent. Findings from this study will undergo peer review and be presented at relevant academic conferences. Any changes to the assessment tools, inclusion criteria, and interventions involved in the implementation of the trial will be immediately reported to the Ethics Committee.

## Trial registration and status

The trial is registered at the Chinese Clinical Trial Registry (ChiCTR2400083643). The study began on 30 April 2024 and continues to enroll participants. The recruitment is expected to be completed by April 30, 2026.

## Discussion

This protocol delineates a structured approach to evaluate the combined effects of RT and VT on the IGF-1/PI3K/AKT/FOXO3 signaling axis in an older population diagnosed with sarcopenia. These cohort patients, characterized by progressive muscle degradation and functional decline, might benefit greatly from advancements in non-pharmacological intervention strategies. Existing studies supported the efficacy of both RT and VT in improving muscle mass and strength in sarcopenia. However, although these benefits are well-documented, the specific molecular mechanisms, particularly the combined impact on muscle anabolic and catabolic pathways, remain unclear. Further research is needed to elucidate how these combined exercises influence key molecular pathways, such as the IGF-1/PI3K/AKT/FOXO3 axis, to enhance therapeutic strategies for sarcopenia.

RT can preserve muscle mass and strength in aging skeletal muscle by activating autophagy and reducing muscle cell apoptosis by modulating the Akt/mTOR and Akt/FOXO3a signaling pathways, as well as IGF-1 and its receptors [62]. Additionally, a study on low-magnitude high-frequency vibration in combination with magnesium supplementation demonstrated a synergistic impact on type I fiber atrophy reduction, significant IGF-1 upregulation, and elevated PI3K/Akt/mTOR signaling, suggesting beneficial effects on sarcopenic muscle [63].

This trial aims to assess the amplified impacts of concurrent RT and VT on the IGF-1/PI3K/AKT/FOXO3 signaling pathway. Understanding these mechanisms could revolutionize sarcopenia care by shifting from isolated exercise modalities to a more integrated therapeutic approach. Although this study follows a rigorous randomized controlled trial design, several limitations should be considered. First, while the study focuses on molecular changes within the IGF-1/PI3K/AKT/FOXO3 axis, it may not capture the full spectrum of physiological adaptations induced by long-term exercise. Second, despite the inclusion of muscle biopsies at baseline, post-intervention (week 12), and follow-up (week 24), the invasive nature of the procedure limited the feasibility of incorporating additional time points. As a result, dynamic signaling fluctuations occurring during the intervention period—particularly acute or transient responses immediately following exercise sessions—may not be fully observed. Third, although participants who had recently taken nutritional supplements were excluded to minimize confounding, we did not formally assess dietary intake throughout the intervention period. This may influence the interpretation of outcomes related to muscle metabolism and function. Fourth, the controlled experimental conditions may not fully reflect the variability of real-world physical activities and lifestyle heterogeneity among older adults, potentially limiting the generalizability of our findings. Future studies should consider incorporating standardized dietary monitoring, employing less invasive sampling methods, or utilizing real-time biomarker tracking to better characterize acute molecular responses and improve ecological validity. Another limitation relates to the design of the control group. Participants in the

control group attended general educational sessions on aging and sarcopenia, which may raise awareness and potentially induce minor behavioral changes. Previous research has shown that even minimal exposure to health information can sometimes influence attitudes toward lifestyle modification. However, studies in elderly populations have consistently reported that education-only interventions do not result in significant improvements in muscle strength, skeletal muscle mass, or functional outcomes [55,56]. Therefore, while we acknowledge that the possibility of small behavioral changes cannot be entirely excluded, the overall risk of substantially underestimating the effects of exercise-based interventions is considered minimal.

Future research should address these limitations by incorporating longitudinal designs to capture the full range of physiological adaptations to long-term training. More frequent muscle biopsies, or less invasive techniques, could provide a comprehensive view of dynamic changes in signaling pathways. A larger and more diverse sample, including individuals with varying levels of baseline physical activity and comorbid conditions, would better represent the broader population with sarcopenia. Expanding research to real-world settings will help validate the findings and improve their applicability to daily activities in older adults.

To enhance the robustness of future interventions, subsequent studies should incorporate longitudinal tracking to assess the durability of the observed effects over extended periods. Expanding the cohort to include more heterogeneous populations would provide insights into differential responses based on genetic, environmental, and lifestyle factors. The integration of continuous, non-invasive biomonitoring technologies could offer a nuanced understanding of real-time biological responses to combined exercise modalities, facilitating personalized adjustment of training protocols. Additionally, exploring the interactions between nutritional status and exercise on the IGF-1/PI3K/AKT/FOXO3 axis could yield critical insights into comprehensive management strategies encompassing both diet and physical activity.

In summary, the outcomes of this study are anticipated to substantially contribute to the body of knowledge regarding the management of sarcopenia through exercise. By delivering empirical evidence on the molecular and functional advantages of combined RT and VT, this study could trigger a paradigm shift in therapeutic strategies designed to improve muscle health and overall quality of life in older populations.

## Supporting information

**S1 File. SPIRIT checklist.**
(DOCX)

**S2 File. Ethical approval document (Chinese and English).**
(PDF)

**S3 File. Study protocol approval (Chinese and English).**
(DOCX)

## Acknowledgments

Not applicable.

## Author contributions

**Formal analysis:** Jianxin Ran, Yan Zheng, Fang Cheng, Yuanpeng Liao.

**Funding acquisition:** Na Li, Jinying Li.

**Methodology:** Jinfeng Yang, Jianxin Ran, Yan Zheng, Yifan Long, Fang Cheng, Yuanpeng Liao.

**Validation:** Yifan Long.

**Visualization:** Jinying Li.

**Writing – original draft:** Haoyang Zhou, Yuanpeng Liao.

**Writing – review & editing:** Jinfeng Yang, Na Li, Yuanpeng Liao.

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
