## [Decision Letter · Decision Letter 0]

15 Jul 2025

Dear Dr. Liao,

Thank you for submitting your manuscript to PLOS ONE. After careful consideration, we feel that it has merit but does not fully meet PLOS ONE’s publication criteria as it currently stands. Therefore, we invite you to submit a revised version of the manuscript that addresses the points raised during the review process.

This paper was considered to be an appropriate study on resistance training, but it needs to be revised based on the comments from the experts. Please provide appropriate responses to the comments from the three reviewers.

Please submit your revised manuscript by Aug 29 2025 11:59PM. If you will need more time than this to complete your revisions, please reply to this message or contact the journal office at plosone@plos.org . A rebuttal letter that responds to each point raised by the academic editor and reviewer(s). You should upload this letter as a separate file labeled 'Response to Reviewers'.A marked-up copy of your manuscript that highlights changes made to the original version. You should upload this as a separate file labeled 'Revised Manuscript with Track Changes'.An unmarked version of your revised paper without tracked changes. You should upload this as a separate file labeled 'Manuscript'.

We look forward to receiving your revised manuscript.

Kind regards,

Tomohiro Yasuda, Ph.D.

Academic Editor

PLOS ONE

2. Thank you for stating the following financial disclosure: [The Institute of Sports Medicine and Health at Chengdu Sports University and the Basic Cultivation Project of Sichuan Provincial Expert Workstation (Grant # SCZJJCC-16) have funded this clinical study.]. 

Reviewers' comments:

Reviewer's Responses to Questions

**Comments to the Author**

1. Does the manuscript provide a valid rationale for the proposed study, with clearly identified and justified research questions?

Reviewer #1: Yes

Reviewer #2: Partly

Reviewer #3: Partly

2. Is the protocol technically sound and planned in a manner that will lead to a meaningful outcome and allow testing the stated hypotheses?

Reviewer #1: Partly

Reviewer #2: Partly

Reviewer #3: Partly

3. Is the methodology feasible and described in sufficient detail to allow the work to be replicable?

Reviewer #1: Yes

Reviewer #2: No

Reviewer #3: No

4. Have the authors described where all data underlying the findings will be made available when the study is complete?

Reviewer #1: No

Reviewer #2: No

Reviewer #3: Yes

5. Is the manuscript presented in an intelligible fashion and written in standard English?

Reviewer #1: Yes

Reviewer #2: Yes

Reviewer #3: Yes

You may also provide optional suggestions and comments to authors that they might find helpful in planning their study.

Reviewer #1: Overall Assessment:

This research protocol aim to investigate the effects of resistance training, vibration training, and their combination on sarcopenia, with a focus on both clinical outcomes and the intracellular IGF-1/PI3K/AKT/FOXO3 signaling axis, addresses a critical gap in the literature. The paucity of human-based evidence concerning these specific molecular pathways in the context of sarcopenia interventions significantly elevates its clinical and academic merit.

Specific Points of Review and Recommendations for Improvement:

1. Methodological Concerns Regarding Molecular Mechanism Elucidation:

The proposed timing of biopsy sample collection appears to be a limiting factor in fully elucidating the intricate molecular mechanisms underpinning the observed clinical effects. Without a more comprehensive sampling strategy (e.g., acute post-intervention time points or multiple time points throughout the intervention period), the study's capacity to precisely delineate the temporal molecular adaptations within the IGF-1/PI3K/AKT/FOXO3 axis remains ambiguous. A more detailed justification for the chosen biopsy time points, or a consideration of additional sampling, is warranted to strengthen the mechanistic insights.

2. Introduction:

Sex-Specific Considerations: The introduction lacks sufficient discussion regarding the known sex differences in sarcopenia prevalence, progression, and responsiveness to exercise interventions. Given the potential for differential responses, it is imperative to integrate relevant literature on sex-specific aspects and outline how these considerations will be addressed within the study design and subsequent analyses. This will enhance the generalizability and clinical relevance of the findings.

Elaboration on Vibration Training (VT) Molecular Mechanisms: The hypothesized molecular mechanisms by which vibration training impacts the IGF-1/PI3K/AKT/FOXO3 axis require more thorough substantiation. While general benefits of VT are often cited, a more detailed review of existing pre-clinical (e.g., animal models) and preliminary human studies that explore the cellular and molecular effects of VT on muscle anabolism and catabolism is necessary. This includes explicitly stating the proposed molecular cascade and formulating clear, testable hypotheses regarding VT's specific influence on the IGF-1/PI3K/AKT/FOXO3 axis.

3. Research Methods:

Design and Grouping – Justification for Intervention and Follow-up Durations: The rationale for the "12-week intervention" and "24-week follow-up" periods is not adequately articulated. A robust justification, grounded in existing literature on exercise physiology, sarcopenia intervention guidelines, or pilot data, is essential to validate these specific durations. This should include considerations for achieving meaningful physiological adaptations and assessing the sustainability of effects.

Sample Size Determination – Consideration of Sex: The current sample size calculation appears to lack consideration for potential sex-specific differences in outcome measures and intervention effects. If significant sex differences are anticipated in sarcopenia progression or response to interventions, the sample size calculation should either account for these variations or explicitly state the implications for generalizability across sexes. Stratification by sex or a power analysis informed by sex-specific variability would enhance the methodological rigor.

Inclusion Criteria – Management of Medication History: The inclusion criteria require more stringent attention to participants' medication history. Pharmacological agents known to influence muscle metabolism, inflammation, or hormonal balance (e.g., corticosteroids, specific cardiovascular medications, diabetes medications, etc.) could confound the study outcomes. A clear strategy for managing these potential confounding factors, whether through explicit exclusion criteria, precise documentation, or statistical adjustment in the analysis, must be elaborated.

Reviewer #2: The primary objective of this 2x2 factorial clinical trial is to evaluate the potential in vivo modulation of molecular mechanisms through combined resistance training (RT) and vibration training (VT) in patients with sarcopenia, and to determine the intervention’s impact on physical function. A total of 96 patients will be randomly allocated to one of four treatment arms: control, RT, VT, and RVT (combined RT and VT), with 24 patients per arm. The intervention period will last 12 weeks, and assessments will be conducted at baseline, after 12 weeks (post-intervention), and at 24 weeks (follow-up). Primary outcome measures include gait speed, handgrip strength, and skeletal muscle mass. Secondary outcome measures encompass IGF-1 concentrations, PI3K/AKT and FOXO3 protein activities, quality of life, and timed-up-and-go test performance. Although the authors discussed the statistical considerations, including the power analysis and sample size estimation, there are several major statistical concerns about this protocol.

Statistical critiques:

1. Although the authors provided a section on sample size determination, the information presented is superficial. The authors should explicitly clarify: (1) the specific statistical tests used for determining the sample size, (2) the anticipated effect sizes among the four treatment arms for each of the three primary endpoints, along with clinical justifications, (3) the identification of the primary measurement time point(s), considering assessments at both 12 and 24 weeks, (4) adjustment methods for the type I error rate, given the multiple comparisons arising from four treatment arms (six possible pairwise comparisons), three primary endpoints, and two assessment time points, and (5) whether an additive or synergistic effect between RT and VT is anticipated, as this significantly influences the power analysis and required sample size.

2. The authors need to clearly state the randomization method employed (e.g., simple randomization, permuted block randomization). Additionally, they should specify if any baseline variables will be used as stratification factors during randomization.

3. It is unclear why the authors did not propose using mixed-effects models for analyzing both primary and secondary outcomes. The authors should clarify their rationale for choosing alternative methods or consider adopting mixed-effects models to accommodate repeated measures and potential within-subject correlations.

4. The authors should explicitly specify the statistical methods intended for calculating 95% confidence intervals (CIs) for all primary and secondary endpoints.

5. The authors should state clearly whether the data analysis will follow an intention-to-treat approach or a modified intention-to-treat methodology.

6. The authors should clearly outline their strategy for multivariable data analysis, addressing the following aspects: (1) model performance evaluation, (2) assumption verification, (3) missing data handling, (4) adjustments for interaction terms and potential nonlinear relationships, and (5) methods planned for internal validation of the analytical model.

7. It is unclear whether the authors have considered including an interim analysis plan.

Reviewer #3: Preventing sarcopenia is a serious worldwide social issue. Resistance exercises are an evidence-based solution to this problem. If there is a way to increase the effectiveness of resistance exercises using a safe and/or low-cost manner, it should be challenged. The proposal in this article, written by Zhou H et al., is interesting from this point of view. However, content revisions and additional information are needed to make this proposal more understandable to the readers.

1) In community-dwelling elderly, the first screening to detect sarcopenia has been effective and recommended for physical function and muscle strength tests. However, loss of muscle mass is the most important indicator in the diagnosis. Therefore, muscle mass should be described before other indicators when describing sarcopenia in an academic article. For example, lines 26-27 (page 3), lines 39-40 (page 3), and line 49 (page 4) should bring the words “size” or “muscle mass” before the others.

2) Lines 50-52: The word “in China” should be included in the text because it is difficult to understand whether the 6.4% in men and 11.5% in women with sarcopenia are in China.

3) Lines 87-88: Correct the sentence as this sentence does not mean “delay the muscle function” but “delay the decline in muscle function”.

4) Lines 114 and 119: “Fig” at the beginning of a sentence should be written as “Figure”.

5) Line 183: Explain in detail the procedures and equipment that the authors plan to use to measure 1 RM.

6) Line 184 and S3 file: The authors plan to use TheraBands in training sessions, but it is unclear how to adjust the training load to 60% or 70% of 1 RM. Explain in detail how to adjust the load so that readers can understand.

7) Line 193: The term “chest depression” is not commonly used in training science. The term should be replaced with “chest press”.

8) Lines 202-203: If the VT consists of 4 sets of 1.5 minutes per session, with 1 minute of interval between the sets, the authors describe the total as 10 minutes (line 199), but isn’t 9 minutes the correct total?

9) Lines 184-185, 207-208: Why is it planned for the RT (70% of the 1 RM for the entire 12-week program) and RVT (50% of 1RM for the first 4 weeks, 55% of 1 RM for the second 4 weeks, and 60% of 1 RM for the final 4 weeks) groups to undergo resistance training at different exercise intensities? Given that the RVT group will be subjected to a lower intensity, any observed difference in training outcomes between the two groups may be confounded by this discrepancy, making it difficult to conclude whether the effects are attributable to training intensity or vibration stimulation. Would it not be more appropriate for both groups to be trained at the same resistance exercise intensity to ensure comparability? If differing training intensities are planned to be employed, the authors should explain scientifically the approach.

10) Lines 214-217: Why do the authors allow the control group to take educational courses? Education can cause behavioral changes in the control group, which may lead to lifestyle changes. Should the RT and RVT groups also take the educational courses? Add a rationalization that no matter what the research results of the study, having only the control group take the educational courses will not interfere with the interpretation of the research results.

11) Line 225: Revise “have sarcopenia” to “have probable sarcopenia” because decreased handgrip strength alone is not sarcopenia, but probable sarcopenia.

12) Lines 220, 228-231: Since the primary outcomes in the study include not only strength and performance (line 220) but also muscle mass (lines 228-231), the content of lines 228-231 should be included at the beginning (line 220).

13) Nutritional intake is also important in sarcopenia, but do the authors not plan to survey it? The authors should explain why the interpretation of the results would not be impaired if they are not done.

14) Revise “Gait speed (>1 m/s)” in Figure 2 to “Gait speed (<1 m/s)”.

15) S3 file, lines 2-3, English page 6: The following sentence is difficult to understand the meaning, “Each movement will be repeated 10 times, with 3 repetitions per set…”. Does “3 repetitions per set” in the sentence mean “3 sets per session”?

**Do you want your identity to be public for this peer review?** For information about this choice, including consent withdrawal, please see our Privacy Policy

Reviewer #1: No

Reviewer #2: No

Reviewer #3: No

---

## [Author Response · Author response to Decision Letter 1]

6 Aug 2025

Dear Editor and Reviewers,

We sincerely appreciate the time and effort that you and the reviewers have devoted to evaluating our manuscript entitled “Impacts of resistance training combined with vibration training on the IGF-1/PI3K/AKT/FOXO3 axis and clinical outcomes in patients with sarcopenia: A protocol for a randomized controlled trial” (Manuscript ID: [PONE-D-25-28564]). We are grateful for the constructive comments and insightful suggestions, which have greatly helped us improve the quality and clarity of the manuscript.

In response to each comment, we have carefully revised the manuscript and provided point-by-point responses below. We hope that the revised version meets the journal’s requirements and addresses all concerns raised during the review process.

Reviewer #1: Overall Assessment:

This research protocol aim to investigate the effects of resistance training, vibration training, and their combination on sarcopenia, with a focus on both clinical outcomes and the intracellular IGF-1/PI3K/AKT/FOXO3 signaling axis, addresses a critical gap in the literature. The paucity of human-based evidence concerning these specific molecular pathways in the context of sarcopenia interventions significantly elevates its clinical and academic merit.

Specific Points of Review and Recommendations for Improvement:

1. Methodological Concerns Regarding Molecular Mechanism Elucidation:

The proposed timing of biopsy sample collection appears to be a limiting factor in fully elucidating the intricate molecular mechanisms underpinning the observed clinical effects. Without a more comprehensive sampling strategy (e.g., acute post-intervention time points or multiple time points throughout the intervention period), the study's capacity to precisely delineate the temporal molecular adaptations within the IGF-1/PI3K/AKT/FOXO3 axis remains ambiguous. A more detailed justification for the chosen biopsy time points, or a consideration of additional sampling, is warranted to strengthen the mechanistic insights.

Reply: We sincerely thank the reviewer for this important comment. To clarify, our current protocol includes muscle biopsies at three time points: baseline (pre-intervention), 12 weeks (post-intervention), and 24 weeks (follow-up). These time points were selected to observe cumulative molecular adaptations before, after, and during the sustained effects following the intervention. We fully agree that more frequent sampling (e.g., acute post-exercise or weekly biopsies) could offer deeper insights into the transient and temporal dynamics of the IGF-1/PI3K/AKT/FOXO3 axis. However, given the invasive nature of muscle biopsy and ethical considerations involving older participants with sarcopenia, we opted to minimize the participant burden and risk by limiting the number of biopsies. In future studies, we plan to explore less invasive approaches such as circulating biomarkers or real-time monitoring techniques, or use animal models to assess rapid molecular responses during intervention. We have added a clarification in the Methods section to explain our rationale and acknowledge this limitation.

2. Introduction:

Sex-Specific Considerations: The introduction lacks sufficient discussion regarding the known sex differences in sarcopenia prevalence, progression, and responsiveness to exercise interventions. Given the potential for differential responses, it is imperative to integrate relevant literature on sex-specific aspects and outline how these considerations will be addressed within the study design and subsequent analyses. This will enhance the generalizability and clinical relevance of the findings.

Reply: We appreciate the reviewer’s thoughtful suggestion regarding the inclusion of sex-specific considerations. We fully agree that sex differences in the sarcopenia prevalence, pathophysiology, and responsiveness to exercise are well-documented and clinically important. To address this, we revised the Introduction section to include additional reference citations on sex-specific patterns of sarcopenia. For instance, studies have shown that women may experience a more rapid decline in muscle strength relative to muscle mass whereas men tend to exhibit more substantial losses in skeletal muscle mass with age. Moreover, differences in hormonal status, fiber-type composition, and neuromuscular control may influence exercise responsiveness according to sex. In the “Statistical analysis” section, we have clarified that sex will be included as a covariate in the ANCOVA and linear mixed models, and exploratory subgroup analyses by sex will also be conducted to identify potential differential effects. We believe that these additions enhance the clinical relevance and generalizability of the study findings.

Elaboration on Vibration Training (VT) Molecular Mechanisms: The hypothesized molecular mechanisms by which vibration training impacts the IGF-1/PI3K/AKT/FOXO3 axis require more thorough substantiation. While general benefits of VT are often cited, a more detailed review of existing pre-clinical (e.g., animal models) and preliminary human studies that explore the cellular and molecular effects of VT on muscle anabolism and catabolism is necessary. This includes explicitly stating the proposed molecular cascade and formulating clear, testable hypotheses regarding VT's specific influence on the IGF-1/PI3K/AKT/FOXO3 axis.

Reply: We thank the reviewer for this excellent comment. We agree that a more detailed explanation of the hypothesized molecular mechanisms of vibration training (VT) is essential for enhancing the theoretical foundation of the study. To address this, we have expanded the Introduction section to include additional evidence from both animal models and preliminary human studies. These studies suggest that VT may stimulate muscle anabolism via the upregulation of IGF-1 expression and downstream activation of the PI3K/AKT signaling cascade, leading to inhibition of FOXO3 nuclear translocation and suppression of proteolytic genes such as MuRF-1 and Atrogin-1. Furthermore, we have clarified the proposed mechanism in our study, as follows. VT may enhance mechanical loading via muscle spindle stimulation and neuromuscular activation, thereby activating mechanosensitive signaling pathways that converge on IGF-1 synthesis. This upregulation is hypothesized to activate PI3K/AKT and suppress FOXO3-mediated muscle degradation. We have now explicitly stated this hypothesis in the revised manuscript and believe it will improve the mechanistic clarity of the protocol.

2. Research Methods:

Design and Grouping – Justification for Intervention and Follow-up Durations: The rationale for the "12-week intervention" and "24-week follow-up" periods is not adequately articulated. A robust justification, grounded in existing literature on exercise physiology, sarcopenia intervention guidelines, or pilot data, is essential to validate these specific durations. This should include considerations for achieving meaningful physiological adaptations and assessing the sustainability of effects.

Reply: We appreciate the reviewer’s thoughtful comment. The selection of a 12-week intervention period was based on extensive literature on exercise physiology and sarcopenia interventions. Previous studies have demonstrated that 12 weeks of resistance or vibration training is sufficient to induce measurable improvements in muscle strength, mass, and hormonal signaling, including IGF-1 pathway activation (Kan et al. 1989; Pereira et al. 2024; Tieland et al. 2012). Specifically, a 12-week duration allows for both neural and hypertrophic adaptations to occur, while maintaining participant compliance in older adults. The 24-week follow-up (i.e., 12 weeks post-intervention) was included to evaluate the sustainability of physiological and molecular adaptations after training cessation. This time frame has been adopted in prior studies on sarcopenia to assess whether benefits in strength and function persist, decline, or plateau in the absence of continued training (Chittrakul et al. 2020). We have now elaborated on these rationales in the “Study design and grouping” section of the revised manuscript.

Sample Size Determination – Consideration of Sex: The current sample size calculation appears to lack consideration for potential sex-specific differences in outcome measures and intervention effects. If significant sex differences are anticipated in sarcopenia progression or response to interventions, the sample size calculation should either account for these variations or explicitly state the implications for generalizability across sexes. Stratification by sex or a power analysis informed by sex-specific variability would enhance the methodological rigor.

Reply: We thank the reviewer for this thoughtful comment. We have revised the “Statistical analysis” section to clarify that sex, along with other key covariates, will be adjusted for using ANCOVA models. Additionally, we will conduct exploratory subgroup analyses stratified by sex and examine sex-by-intervention interaction terms to evaluate potential moderation effects. Although the current sample size is not specifically powered for interaction effects, these analyses will help identify sex-specific trends and guide future trials. This information has been added to the revised manuscript (Methods > Statistical analysis)

Inclusion Criteria – Management of Medication History: The inclusion criteria require more stringent attention to participants' medication history. Pharmacological agents known to influence muscle metabolism, inflammation, or hormonal balance (e.g., corticosteroids, specific cardiovascular medications, diabetes medications, etc.) could confound the study outcomes. A clear strategy for managing these potential confounding factors, whether through explicit exclusion criteria, precise documentation, or statistical adjustment in the analysis, must be elaborated.

Reply: We appreciate the reviewer’s valuable recommendation. In response, we have revised the “Exclusion criteria” section to explicitly exclude participants who are currently receiving medications that are known to substantially affect muscle metabolism or hormonal regulation, including systemic corticosteroids, anabolic agents, or hormone replacement therapy. Additionally, the use of medications for cardiovascular or metabolic conditions (e.g., beta-blockers, statins, insulin sensitizers) will be carefully documented during screening and baseline assessment. These variables will be considered covariates in the statistical analysis if they show imbalance between groups or correlations with outcome measures.

Reviewer #2: The primary objective of this 2x2 factorial clinical trial is to evaluate the potential in vivo modulation of molecular mechanisms through combined resistance training (RT) and vibration training (VT) in patients with sarcopenia, and to determine the intervention’s impact on physical function. A total of 96 patients will be randomly allocated to one of four treatment arms: control, RT, VT, and RVT (combined RT and VT), with 24 patients per arm. The intervention period will last 12 weeks, and assessments will be conducted at baseline, after 12 weeks (post-intervention), and at 24 weeks (follow-up). Primary outcome measures include gait speed, handgrip strength, and skeletal muscle mass. Secondary outcome measures encompass IGF-1 concentrations, PI3K/AKT and FOXO3 protein activities, quality of life, and timed-up-and-go test performance. Although the authors discussed the statistical considerations, including the power analysis and sample size estimation, there are several major statistical concerns about this protocol.

Statistical critiques:

1. Although the authors provided a section on sample size determination, the information presented is superficial. The authors should explicitly clarify: (1) the specific statistical tests used for determining the sample size, (2) the anticipated effect sizes among the four treatment arms for each of the three primary endpoints, along with clinical justifications, (3) the identification of the primary measurement time point(s), considering assessments at both 12 and 24 weeks, (4) adjustment methods for the type I error rate, given the multiple comparisons arising from four treatment arms (six possible pairwise comparisons), three primary endpoints, and two assessment time points, and (5) whether an additive or synergistic effect between RT and VT is anticipated, as this significantly influences the power analysis and required sample size.

Reply: We thank the reviewer for this detailed and insightful comment. We have revised the “Sample size determination” section to address each of the concerns listed below.

(1) The sample size was calculated based on two-way repeated-measures ANOVA with interaction, using G*Power 3.1.

(2) Anticipated effect sizes (partial η2 = 0.14, corresponding to Cohen's f = 0.40) were derived from previous studies examining the effects of resistance and vibration training on IGF-1 expression and muscle function in older adults. These represent moderate-to-large clinically meaningful changes.

(3) The primary endpoint is the change from baseline to 12 weeks, which aligns with the end of the supervised intervention and is expected to capture peak molecular and functional adaptations. The 24-week point serves as a follow-up to assess sustainability but was not used in powering the study.

(4) Although multiple endpoints and comparisons exist, we acknowledge the inflation of type I error. Therefore, Holm–Bonferroni correction will be applied in the main analysis; this has been reflected in the power calculation assumptions.

(5) We hypothesize an additive effect of RT and VT, rather than synergistic interaction. Thus, no interaction effect was assumed in the initial power analysis. Future analyses will explore potential synergy via interaction terms, but the current sample size ensures detection of additive group differences.

The revised paragraph is now included in the “Sample size determination” section of the Methods.

2. The authors need to clearly state the randomization method employed (e.g., simple randomization, permuted block randomization). Additionally, they should specify if any baseline variables will be used as stratification factors during randomization.

Reply: We appreciate the reviewer’s important observation. We have clarified the randomization method in the “Randomization and blinding” section. Specifically, a permuted block randomization (block size of 4) will be used to ensure balanced group sizes. Randomization will be conducted by an independent statistician using a computer-generated random sequence. No stratification by baseline variables (e.g., age, sex) will be used in the randomization process. However, these variables will be included as covariates in the statistical analysis to control for potential confounding.

3. It is unclear why the authors did not propose using mixed-effects models for analyzing both primary and secondary outcomes. The authors should clarify their rationale for choosing alternative methods or consider adopting mixed-effects models to accommodate repeated measures and potential within-subject correlations.

Reply: Thank you for your suggestion. In the revised manuscript, we clarified that two-way repeated-measures ANOVA will be used for primary outcomes owing to the balanced design. However, we have now specified the use of linear mixed-effects models for secondary longitudinal outcomes (e.g., gait speed), which can better accommodate within-subject correlations and missing data.

4. The authors should explicitly specify the statistical methods intended for calculating 95% confidence intervals (CIs) for all primary and secondary endpoints.

Reply: We appreciate the reviewer’s valuable comment. In the revised manuscript, we have now explicitly stated that 95% confidence intervals for all effect sizes (Cohen’s d) and model estimates (e.g., from linear mixed models) will be computed using bias-corrected and accelerated (BC) bootstrap methods or model-based standard errors, depending on the test applied.

5. The authors should state clearly whether the data analysis will follow an intention-to-treat approach or a modified intention-to-treat metho

---

## [Decision Letter · Decision Letter 1]

19 Aug 2025

Dear Dr. Liao,

Thank you for submitting your manuscript to PLOS ONE. After careful consideration, we feel that it has merit but does not fully meet PLOS ONE’s publication criteria as it currently stands. Therefore, we invite you to submit a revised version of the manuscript that addresses the points raised during the review process.

**Your research paper is currently being reviewed for publication, but it still needs further improvement. Please respond to the reviewers' comments appropriately**
**.**

We look forward to receiving your revised manuscript.

Kind regards,

Tomohiro Yasuda, Ph.D.

Academic Editor

PLOS ONE

Journal Requirements:

Reviewers' comments:

Reviewer's Responses to Questions

**Comments to the Author**

1. Does the manuscript provide a valid rationale for the proposed study, with clearly identified and justified research questions?

Reviewer #1: Yes

Reviewer #2: Yes

Reviewer #3: Yes

2. Is the protocol technically sound and planned in a manner that will lead to a meaningful outcome and allow testing the stated hypotheses?

Reviewer #1: Yes

Reviewer #2: No

Reviewer #3: Yes

3. Is the methodology feasible and described in sufficient detail to allow the work to be replicable?

Reviewer #1: Yes

Reviewer #2: Yes

Reviewer #3: No

4. Have the authors described where all data underlying the findings will be made available when the study is complete?

Reviewer #1: Yes

Reviewer #2: Yes

Reviewer #3: Yes

5. Is the manuscript presented in an intelligible fashion and written in standard English?

Reviewer #1: Yes

Reviewer #2: Yes

Reviewer #3: Yes

You may also provide optional suggestions and comments to authors that they might find helpful in planning their study.

Reviewer #1: The authors have successfully addressed all of my previous comments. The manuscript is much improved, and I have no further concerns at this time.

Reviewer #2: The manuscript is much improved.

Reviewer #3: The revised manuscript by Zhou H et al. almost responded to my questions. However, there is one point that needs to be revised.

1) The authors revised that the educational sessions for the control group will not influence behaviors related to daily physical activity levels or nutritional intake in elderly people (lines 312 to 318). Many studies have reported that posters encouraging people of various ages to quit smoking are effective in promoting smoking cessation, and that posters encouraging exercise increase the physical activity levels people engage in. In this way, it has been shown that providing even a small amount of information can lead to behavioral change. Thus, it is difficult to interpret that the educational sessions for the elderly people, which consisted of two 60-minute classes per month on sarcopenia and health, will not promote behavioral change in elderly people. Previous studies showing that such educational interventions do not promote behavioral change, or even if they do, they do not result in improved muscle strength or physical function, should be presented. It is important to carefully evaluate the effects of educational sessions, as otherwise there is a risk of underestimating the value of physical training in other groups.

**Do you want your identity to be public for this peer review?** For information about this choice, including consent withdrawal, please see our Privacy Policy

Reviewer #1: No

Reviewer #2: No

Reviewer #3: No

---

## [Author Response · Author response to Decision Letter 2]

20 Aug 2025

Dear Editor and Reviewers,

We sincerely thank you for your time, effort, and constructive feedback on our manuscript entitled “Impacts of resistance training combined with vibration training on the IGF-1/PI3K/AKT/FOXO3 axis and clinical outcomes in patients with sarcopenia: A protocol for a randomized controlled trial” (Manuscript ID: PONE-D-25-28564R1).

We are grateful for the valuable comments and suggestions provided in both the first and second rounds of review, which have greatly helped us improve the clarity, rigor, and overall quality of our work. We are particularly encouraged by the reviewers’ recognition that the revised manuscript has been substantially improved.

In this revised version, we have carefully addressed all the remaining concerns raised by the reviewers. Specifically, we further clarified the rationale and potential influence of the educational sessions in the control group, and we added relevant references to support our explanation. All changes have been highlighted in the revised manuscript for ease of review.

We once again thank the reviewers and the editor for their insightful comments and guidance, which have significantly strengthened our manuscript. We hope that our revisions meet with your approval.

Reviewer #1: The authors have successfully addressed all of my previous comments. The manuscript is much improved, and I have no further concerns at this time.

Response to Reviewer #1: We sincerely appreciate your positive feedback and recognition of our revisions. We are very grateful for your constructive comments in the first round, which greatly helped us improve the quality of the manuscript. Thank you again for your time and support.

Reviewer #2: The manuscript is much improved.

Response to Reviewer #2: We sincerely thank you for your positive evaluation of our revised manuscript. We greatly appreciate your comments and recognition, which encouraged us to further refine and improve our work. Thank you again for your valuable time and support.

Reviewer #3: The revised manuscript by Zhou H et al. almost responded to my questions. However, there is one point that needs to be revised.

1) The authors revised that the educational sessions for the control group will not influence behaviors related to daily physical activity levels or nutritional intake in elderly people (lines 312 to 318). Many studies have reported that posters encouraging people of various ages to quit smoking are effective in promoting smoking cessation, and that posters encouraging exercise increase the physical activity levels people engage in. In this way, it has been shown that providing even a small amount of information can lead to behavioral change. Thus, it is difficult to interpret that the educational sessions for the elderly people, which consisted of two 60-minute classes per month on sarcopenia and health, will not promote behavioral change in elderly people. Previous studies showing that such educational interventions do not promote behavioral change, or even if they do, they do not result in improved muscle strength or physical function, should be presented. It is important to carefully evaluate the effects of educational sessions, as otherwise there is a risk of underestimating the value of physical training in other groups.

Response to Reviewer #3: We sincerely thank you for this important comment. We fully agree with your concern that educational interventions may induce behavioral changes in elderly people, as suggested by previous evidence from public health campaigns. To address this, we have carefully revised the manuscript:

In the Methods – Educational courses section, we clarified that the educational sessions were designed as general health information lectures, standardized and non-interactive, with no specific guidance on exercise practice or nutrition planning.

We have now added references to previous studies reporting that educational lectures alone did not lead to significant improvements in muscle strength, mass, or functional outcomes in elderly populations with sarcopenia or frailty.

In the Discussion section, we further acknowledged that although minor behavioral changes cannot be excluded, prior evidence indicates that education-only interventions rarely translate into measurable improvements in physical performance. Therefore, the risk of overestimating the effects in the control group is minimal.

We sincerely appreciate your suggestion, which helped us refine the rationale of our control group design and avoid potential underestimation of the exercise interventions.

---

## [Decision Letter · Decision Letter 2]

15 Sep 2025

Impacts of resistance training combined with vibration training on the IGF-1/PI3K/AKT/FOXO3 axis and clinical outcomes in patients with sarcopenia: A protocol for a randomized controlled trial

PONE-D-25-28564R2

Dear Dr. Yuanpeng Liao,

We’re pleased to inform you that your manuscript has been judged scientifically suitable for publication and will be formally accepted for publication once it meets all outstanding technical requirements.

Kind regards,

Tomohiro Yasuda, Ph.D.

Academic Editor

PLOS ONE

Additional Editor Comments (optional):

Reviewer #2: None

Reviewer #3: None

Reviewers' comments:

Reviewer's Responses to Questions

**Comments to the Author**

1. Does the manuscript provide a valid rationale for the proposed study, with clearly identified and justified research questions?

Reviewer #2: Yes

Reviewer #3: Yes

2. Is the protocol technically sound and planned in a manner that will lead to a meaningful outcome and allow testing the stated hypotheses?

Reviewer #2: Yes

Reviewer #3: Yes

3. Is the methodology feasible and described in sufficient detail to allow the work to be replicable?

Reviewer #2: Yes

Reviewer #3: Yes

4. Have the authors described where all data underlying the findings will be made available when the study is complete?

Reviewer #2: Yes

Reviewer #3: Yes

5. Is the manuscript presented in an intelligible fashion and written in standard English?

Reviewer #2: Yes

Reviewer #3: Yes

You may also provide optional suggestions and comments to authors that they might find helpful in planning their study.

Reviewer #2: The authors have responded well to the statistical issues raised in the previous review. There is no further statistical concern about this revised manuscript.

Reviewer #3: The revised manuscript adequately addresses all points of concern and is acceptable. In my opinion, this article is worthy of publication in the PLoS ONE.

**Do you want your identity to be public for this peer review?** For information about this choice, including consent withdrawal, please see our Privacy Policy

Reviewer #2: No

Reviewer #3: No

---

## [Editor Report · Acceptance letter]

PONE-D-25-28564R2

PLOS ONE

Dear Dr. Liao,

I'm pleased to inform you that your manuscript has been deemed suitable for publication in PLOS ONE. Congratulations! Your manuscript is now being handed over to our production team.

Kind regards,

on behalf of

Prof. Tomohiro Yasuda

Academic Editor

PLOS ONE